# Congenital Cutis Verticis Gyrata in a Newborn with Turner Syndrome: A Rare Clinical Manifestation of This Chromosomal Disease with Trichoscopic Evaluation

**DOI:** 10.3390/diagnostics13152574

**Published:** 2023-08-02

**Authors:** Riccardo Bortone, Domenico Bonamonte, Gerardo Cazzato, Carmelo Laface, Alberto Gaeta, Teresa Lettini, Caterina Foti, Raffaele Filotico, Francesca Ambrogio

**Affiliations:** 1Section of Dermatology and Venereology, Department of Precision and Regenerative Medicine and Ionian Area (DiMePRe-J), University of Bari “Aldo Moro”, 70124 Bari, Italy; riccardobort@gmail.com (R.B.); domenico.bonamonte@uniba.it (D.B.); teresa.lettini@uniba.it (T.L.); caterina.foti@uniba.it (C.F.); raffaele.filotico@uniba.it (R.F.); dottambrogiofrancesca@gmail.com (F.A.); 2Section of Molecular Pathology, Department of Precision and Regenerative Medicine and Ionian Area (DiMePRe-J), University of Bari “Aldo Moro”, 70124 Bari, Italy; 3Medical Oncology, Dario Camberlingo Hospital, 72021 Francavilla Fontana, Italy; carmelo.laface@gmail.com; 4Radiology Unit, Pediatric Hospital Giovanni XXIII, 70126 Bari, Italy; al.gaeta@libera.it

**Keywords:** congenital cutis verticis gyrate, Turner syndrome, chromosomal disease, trichoscopy

## Abstract

Cutis verticis gyrata (CVG) is a rare disorder of the scalp that entails the development of ridges and furrows, which mimic the anatomical conformation of the brain. This skin condition has been classified in primary essential, primary non-essential, and secondary CVG, depending on the presence or absence of other associated disorders. We present the case report of a one-month-old female newborn affected by congenital CVG (CCVG), who also received a diagnosis of Turner syndrome (TS). Skin folding was present at birth and located at the left frontal region of the scalp in the sagittal plane. Our purpose was to make this pathology clinically and tricoscopically better known, since it can be related to different genetic, inflammatory, and neoplastic conditions, etc. Non-invasive investigations, such as ultrasonography (U/S) of the brain and scalp and trichoscopy, were also used to obtain the important clues necessary to help in the CVG classification. The clinical diagnosis and trichoscopical investigation of CVG may also be useful for those patients who may have a genetic disease that is not screened for during prenatal examinations.

Cutis verticis gyrata (CVG) is a rare disorder of the scalp that entails the development of ridges and furrows, which mimic the anatomical conformation of the brain.

**Figure 1 diagnostics-13-02574-f001:**
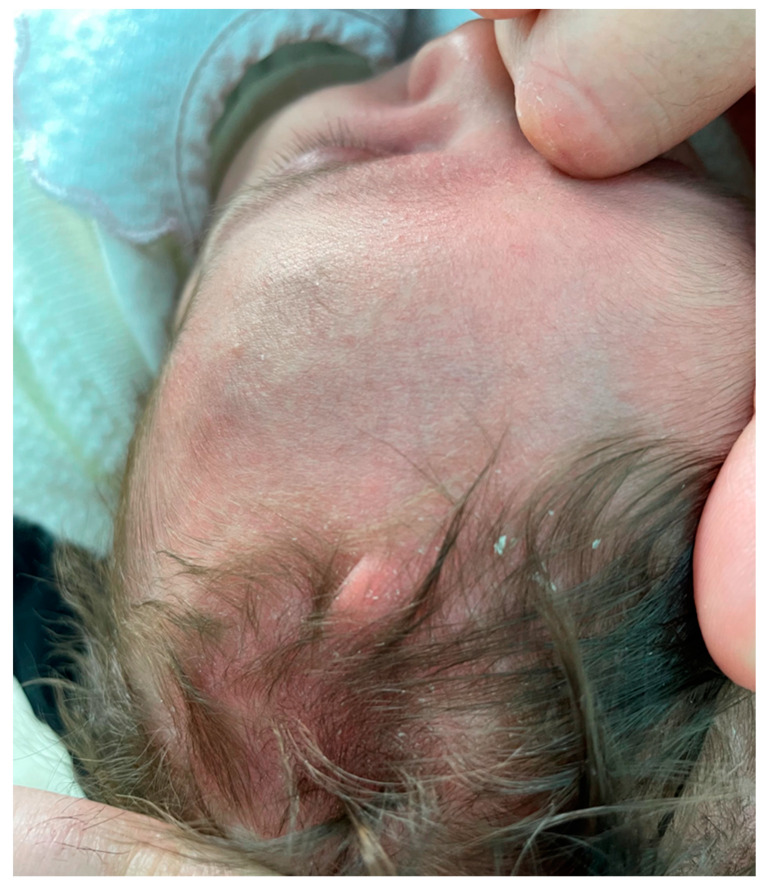
Raised, spongy, fleshy colored skin folding with few hairs’ growth above it. Encephalic ultrasonography (U/S) was performed to exclude any correlated disorders underneath the scalp lesion. Additionally, U/S of the alteration was carried out to understand the general features of the lesion. Trichoscopy and trichogram were performed to obtain microscopic findings. We present a case report of a one-month-old female newborn affected by congenital CVG (CCVG) with a concomitant diagnosis of Turner syndrome (TS). The latter was made at pregnancy week 12 by chorionic villus sampling because of an increase in nuchal translucency observed using sonography. Skin folding was present at birth and located at the left frontal region of the head in the sagittal plane. It was raised, spongy, fleshy colored, and with few hairs’ growth above it (Figure 1). Our purpose is to describe a further case of this rare scalp manifestation associated with TS also by using a trichoscopical investigation.

**Figure 2 diagnostics-13-02574-f002:**
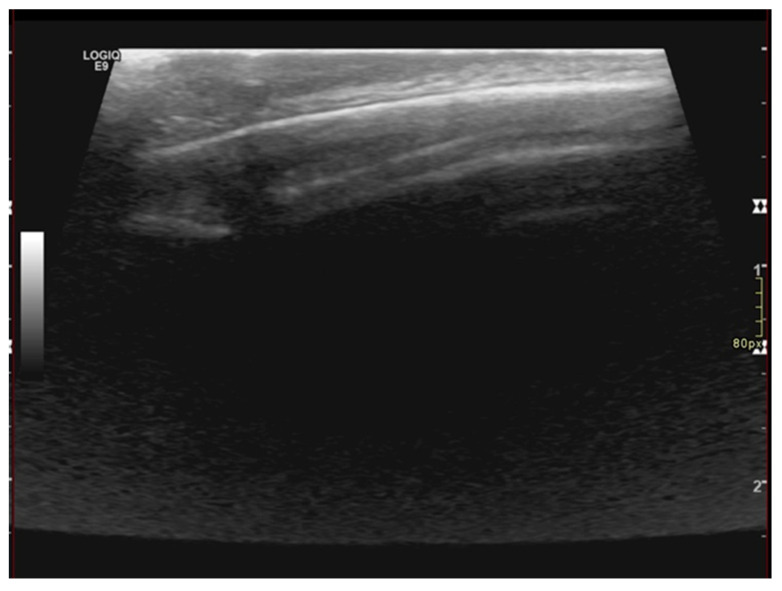
U/S of the scalp lesion showing thickening epicranial tissues without alteration of the echostructure and absence of focal lesions. The thickness of the skin fold is 6 mm. Brain U/S investigation did not reveal any abnormality of the posterior cranial fossa and of the encephalic tissue. U/S of the skin fold and scalp showed regular soft tissue appearance and normal bony structures of the skull. Performing U/S of the scalp, we found thickening of the epicranial tissues without alteration of the echostructure and absence of focal lesions. In addition, the measured thickness of the skin fold was 6 mm (Figure 2).

**Figure 3 diagnostics-13-02574-f003:**
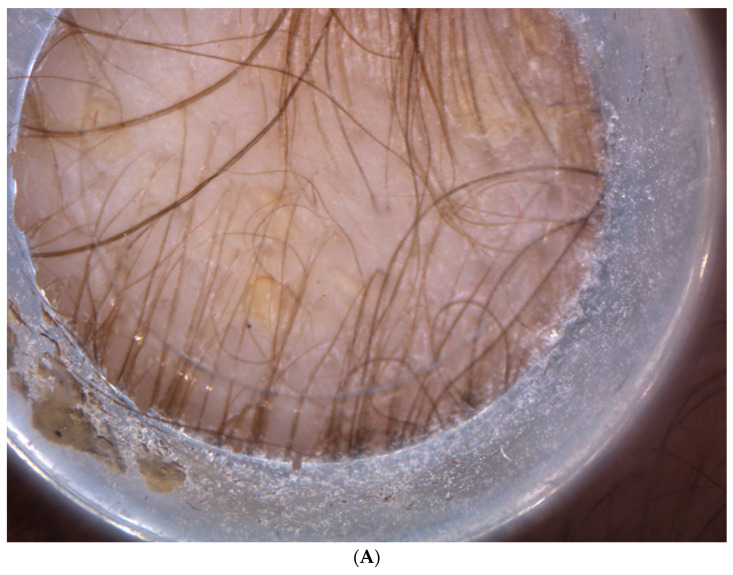
Higher magnification of the skin folding conducted using Vidix video dermatoscopy (**A**,**B**) Absence of abnormalities of the hair shaft as pili torti, minimal non-homogeneous thickness of the hair with caliber variation, decreased hair density, and scaly patches due to cradle cap. The trichoscopic studies of the ridge demonstrated an absence of scarring alopecia. From the analysis, a minimal non-homogeneous thickness of the hair with caliber variation was disclosed. Moreover, further findings revealed a decreased hair density, the absence of yellow dots, and the presence of scaly patches due to cradle cap, as shown in the higher magnification pictures obtained using Vidix video dermatoscopy (Figure 3A,B).

**Figure 4 diagnostics-13-02574-f004:**
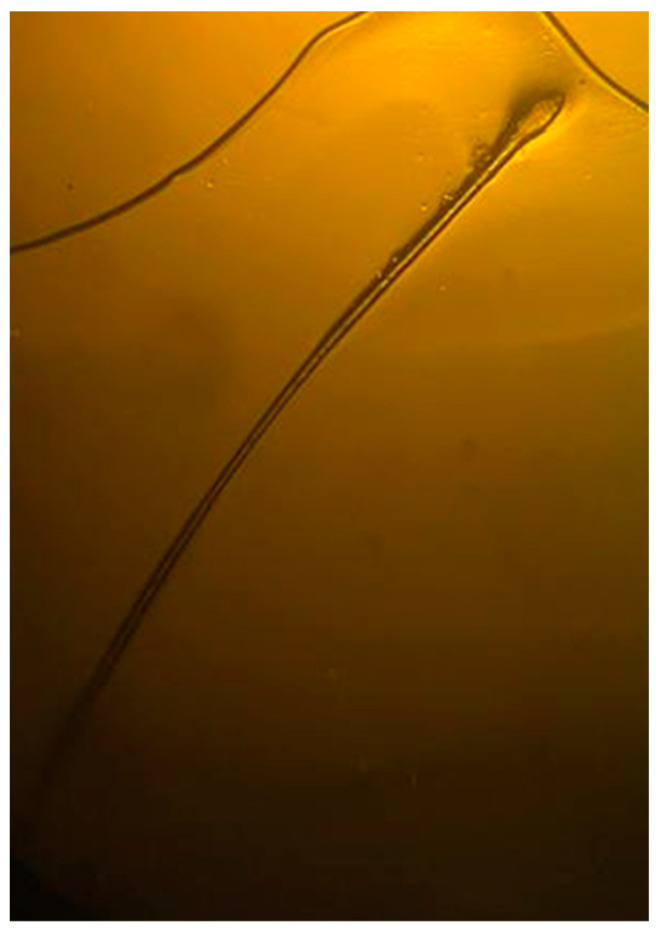
Trichogram of the hair located at the skin lesion showed absence of abnormalities of the shaft and physiological telogen phase. The trichogram of the hair located at the skin lesion showed the absence of abnormalities of the shaft, such as pili torti (Figure 4). Moreover, the tested hair was in a physiological telogen phase (Figure 4), without the presence of dystrophic hair. Finally, any change in the color of the skin compared to the surrounding normal scalp was not divulged.

CVG is a skin proliferation that has the appearance of the sulci and gyri of the brain [1,2]. It is a rare condition with a stated prevalence of about 1 in 100,000 males and 0.026 in 100,000 females [3].

According to the literature, resolving lymphedema in utero may be the cause of this rare skin presentation [4]. It has been classified in primary essential, primary non-essential, and secondary CVG, depending on the presence or absence of other associated abnormalities. Primary essential CVG does not involve any other underlying anomaly besides hypertrophy of the connective tissue. Primary non-essential CVG is related to neuropsychiatric (intellectual disabilities, developmental delay, cerebral palsy, epilepsy, schizophrenia, and microcephaly) and ophthalmologic disorders (cataract, strabismus, blindness, and retinitis pigmentosa) [5]. Secondary CVG is due to several conditions such as an inflammatory process (eczema, psoriasis, folliculitis, impetigo, erysipelas, pemphigus, acne keloidalis, and acne inversa), hamartoma of the scalp (congenital or acquired intradermal melanocytic naevus, sebaceous naevus, naevus lipomatosus, connective tissue naevus, and neurofibroma), systemic diseases (myxoedema, acanthosis nigricans, acromegaly, diabetes mellitus, the misuse of anabolic substances, amyloidosis, syphilis, chronic pulmonary diseases, congenital cyanotic heart disease, hepatobiliary disorders, leukaemia, and paraneoplastic syndromes), cancers, or drugs [3,5]. In the primary types of CVG, the skin fold runs antero-posteriorly in the sagittal plane of the head, while in secondary CVG, it does not present in the same direction [5]. CCVG has been reported in association with some genetic disorders such as TS, Klinefelter syndrome, Noonan syndrome, fragile chromosome syndromes, Beare–Stevenson cutis gyrata syndrome, Ehlers–Danlos syndrome, Apert syndrome, and tuberous sclerosis [3,5,6]. Our case report describes a very rare case of CCVG associated with TS. According to the literature, 13 other cases have been reported with the same association, but only two cases had frontal localization [7]. We performed non-invasive investigations such as brain and skin fold sonographic studies that did not show any associated tissue anomalies under the scalp lesion, which excluded the secondary form. 

Furthermore, the trichoscopic analysis disclosed a slight non-homogeneous thickness of the hair with caliber variation compared to the surrounding scalp, but without the signs of scarring alopecia that may be seen in the secondary types of CVG due to its underlying process [5]. Noticeably, regarding the secondary forms of CVG, there have been few reported cases of TS patients with CVG related to dermal hamartomas [8]. These studies revealed that tumor-like lesions such as hamartomas may have an increased amount of collagen and mucin on histology; thus, these features may affect proper hair growth, and this may be detected by trichoscopy [5,7]. Hamartoma is benign condition that may occur in different sites of the body. Concerning our discussion, it is included in the causes of secondary CVG. Histopathology studies documented the presence of increased mucin deposition, connective tissue hypertrophy, a thick fibrous band of collagen in the reticular and fascicular dermis, and the proliferation of collagen fibers on the papillary and reticular dermis [7]. Thus, some CVG may be dermal hamartoma and therefore this represents a secondary form of CVG. This may be concluded via a biopsy, which we could not perform because of the parents’ decision. Moreover, another cause of congenital secondary CVG could also be a hamartomatous lesion such as an intradermal melanocytic naevus [5]. In this circumstance, trichoscopy may help to evaluate the presence or absence of pigment patterns and rule out the presence of a melanocytic lesion. In this way, trichoscopy may be useful for dermatologists who examinate the scalp lesions of Turner syndrome patients to quickly orientate the diagnosis of CVG. 

Regarding our results, considering the absence of hypotonia and the impossibility of evaluating any kind of developmental disability because of the patient’s age, we may suppose that our case of CCVG could be classified as primary essential CVG related to TS with frontal localization, which should be the third case in the literature. A biopsy of the lesion and magnetic resonance imaging should also have been carried out to conclude the work up, but the parents denied it. 

According to the literature, the histopathologic features of primary CVG range from a normal skin structure to thickened connective tissue with hypertrophy or hyperplasia of adnexal structures. Instead, the histopathologic findings in secondary CVG depend on the underlying condition [2]. In a case report about a secondary CVG located at the glabella, the authors documented the presence of a thickened dermis, the prominence of the pilosebaceous unit, the sweat gland ducts extending focally into the subcutis, and chronic inflammation with plasma cells and eosinophils [9]. 

Finally, the presence of an underlying genetic disease which may have been associated and that is not routinely screened for could be hypothesized from the clinical and trichoscopic diagnosis of CVG. Performing trichoscopy could help to distinguish the primary type of CVG from the secondary one by the presence or absence of scarring alopecia and other features of the underlying hamartoma, such as pigment. In that occurrence, this may suggest the presence of a melanocytic lesion of the scalp.

Our work aims to make this pathology better known clinically and trichoscopically so it can be more easily diagnosed and to encourage researchers to further study the association with the chromosomal anomaly. Moreover, by presenting this case, we want to demonstrate how non-invasive methods may help to understand the main features of skin lesions and to rule out anatomical changes below them. Additionally, trichogram and trichoscopical studies may give important clues to help in the classification of lesions, even though further studies are needed for this scope.

## Data Availability

Not applicable.

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
