# Peer review of "Congenital Cutis Verticis Gyrata in a Newborn with Turner Syndrome: A Rare Clinical Manifestation of This Chromosomal Disease with Trichoscopic Evaluation"

_diagnostics, 2023, doi:10.3390/diagnostics13152574_

Round 1

Reviewer 1 Report

Dear authors, thanks for your very interesting case report.

Some minor queries:

Legend of fig. 2 "with normal thickness on the skin folding" - how could measure the skin thickness by trichoscopy?

p. 4: "paraneoplastic syndromes) cancers or drugs" - please insert a comma before cancers.

p. 4: "Furthermore, trichoscopic analysis disclosed no signs of alopecia" - Yes but you may mention the decreased hair density.

Additional question: Do you have a higher magnification picture of trichoscopy?

Author Response

Reviewer n'1: How could measure the skin thickness by trichoscopy?

With trichoscopy we could measure the skin thickness of the lesion, but we performed ultrasound of the scalp and we found that the thickness of the skin lesion was 6mm. In the article we added what we could appreciate by sonography and we attached U/S pictures.

2. We have inserted comma before cancers and mentioned decreased hair density.

  1. Do you have a higher magnification picture of trichoscopy?

We added Figures 4 A and B in the text.

Reviewer 2 Report

The authors present a case report of CCVG in the rare association with TS. In the cited literature, another 13 cases have been reported with the same association, two of them with an identical frontal localization. 

Without giving any relevance, the authors hypothesize that trichoscopy may help to distinguish the primary type of CVG from the secondary type by the presence or absence of alopecia and that a biopsy of the lesion and magnetic resonance imaging should be carried out to conclude the work up. In contrast to these statements, the case report does not show any trichoscopic details, nor any microscopic details of a skin biopsy or the results of MRI. 

In summary, the research is not conducted correctly and additional investigations are needed

Minor editing of English language is required. 

Author Response

Dear Sir/Madame,

We are aware that biopsy and magnetic resonance imaging should have been carried out, but patient’s parents denied it and thus we could not perform them.

Despite all of this, we now provide higher magnification pictures of the trichoscopy of the lesion to have a better view of it and the sonographic imaging that shows the thickness of the fold at the cutaneous and subcutaneous level.

By adding the clinical, trichoscopic and ultrasonographic description of this rare manifestation of Turner Syndrome, we hope to have the chance to give our contribute to the literature about this uncommon skin condition.

Reviewer 3 Report

If performed what coud yu expect from histopathology? Do you know some example of trycoscopic and histologic correlations?What about a theory of hamartoma?

Author Response

  1. If performed, what could you expect from histopathology?

As reported by Reference 2, histopathologic features of primary CVG range from normal skin structure to thickened connective tissue with hypertrophy or hyperplasia of adnexal structures. Instead, histopathologic findings in secondary CVG depends on the underlying condition. For instance, in case report about a secondary CVG located at glabella, presented by Harish et Clark, they documented thickened dermis, prominence of the pilosebaceous unit and sweat gland ducts extending focally into subcutis. Chronic inflammation with plasma cells and eosinophils were also noted (9).

  1. Do you know some example of trichoscopy and histologic correlations?

As reported by References 5 and 8, in primary CVG the texture of the hair is normal and there is not a significant decrease in hair density. On the other hand, tumor-like lesions may show progressive alopecia. According to this, benign tumor-like conditions as hamartomas, may have an increased amount of collagen and mucin on histology and this may affect the proper hair growth, and this may be detected by trichoscopy.  

Moreover, congenital secondary CVG may also be caused by hamartomatous lesions such as an intradermal melanocytic naevus (Reference 5). In this circumstance, trichoscopy may help to evaluate the presence or absence of pigment patterns and to rule out the presence of a melanocytic lesion. In this way, trichoscopy may be useful for dermatologists who examinate a scalp lesion of a Turner syndrome patient to quickly orientate the diagnosis of CVG.

  1. What about the theory of hamartoma?

Hamartoma is benign tumor-like mass that may occur in different sites of the body.

Regarding our topic, it is included in the causes of secondary CVG. Histopathology studies of Corona et al. (Reference 8) documented the presence of increase mucin deposition, connective tissue hypertrophy, the thick fibrous band of collagen in the reticular and fascicular dermis and the proliferation of collagen fibers on the papillary and reticular dermis. In conclusion, some CVG may be dermal hamartoma and thus a secondary form of CVG. This may be achieved by biopsy that we could not perform because of parents’ decision.

Round 2

Reviewer 2 Report

The type of trichoscopy that the authors have applied in Figures 3 and 4 represents a simple, undemanding high magnification reflected light microscopy of the hair.

The authors claim, Figure 3 “shows few hairs scattered with normal thickness on the skin folding”. According to my impression, the hair on the head is light and thin instead. 

The statement, Fig. 4A represents a higher magnification of Figure 3, is not recognizable for me, at least it shows a different section. 

The statement, Fig. 4A shows absence of abnormalities of the hair shaft, is incomprehensible for me. On the contrary, I see dystrophic hair next to atrophic ones as well as caliber variation of hair in addition to follicular keratoses in Fig. 4A. 

Furthermore, the description on Fig. 4B that the thickness of hair existing at the lesion is the same as that of the hair present at adjacent scalp and without significant decreased hair density is not visible to me. 

These uncertainties in the present trichoscopic investigation could have been avoided by performing a classic trichogram of cut hair or refering to the likewise non-invasive digital analysis of hair, in which the hair density and the condition of the hair shafts are representatively and repeatably determined by scanning electron microscopy or with the aid of a computer by video-dermatoscopy. 

In line 63 it reads "few hairs" instead of "few hair". 

Author Response

Bari (Italy), 28th June 2023

Dear Reviewer,

we appreciate your careful interest to our manuscript.

We checked the Figure 3 that we performed by DermLite IV DL4 Polarized Dermatoscope 30mm field of view with 10x magnification may be confusing, so we decided to delete it. Indeed, by figure 3 we cannot appreciate the proper thickness of the hair at the lesion. Instead, by Figure 4 performed by Vidix video-dermatoscope, we can appreciate that the hair at the skin fold present a minimal non-homogeneous thickness with caliber variation. Moreover, we highlight a decreased hair density, absence of yellow dots and presence of scaly patches due to cradle cap,

Regarding figure 4A-4B, when we speak about abnormalities of the hair shaft, we meant the absence of pili torti. We cannot appreciate atrophic and dystrophic hair because trichogram disclosed that hair were in a telogen physiological phase as shown by the new figure 4 that we insert in the new text and below here.

Figure 4: Trichogram of the hair located at the skin lesion shown absence of abnormalities of the shaft and physiological telogen phase.

Kind regards

Round 3

Reviewer 2 Report

With respect to Figure 3, it is difficult to understand that the authors delete it after former objections and with the statement, they „cannot appreciate the proper thickness of the hair at the lesion“. In Figure 4 the authors confirm the objection, "the hair at the skin fold present a .… caliber variation“ and „…. a decreased hair density". These severe modifications contradict the original description in the first submitted report.

The claim, „abnormalities of the hair shaft“ meant "the absence of pili torti“ is incomprehensible because nonsensical.

The claim „We cannot appreciate atrophic and dystrophic hair because trichogram disclosed that hair“ is not convincing because the single hair in the newly inserted Figure 4 is not representative for a trichogram but displays one single hair shaft with a hair root in telogen stage. Whether this is in context to the findings in the former Figure 4 remains debatable.